# Mechanisms of Melanoma Progression and Treatment Resistance: Role of Cancer Stem-like Cells

**DOI:** 10.3390/cancers16020470

**Published:** 2024-01-22

**Authors:** Youssef Al Hmada, Robert T. Brodell, Naji Kharouf, Thomas W. Flanagan, Abdulhadi A. Alamodi, Sofie-Yasmin Hassan, Hosam Shalaby, Sarah-Lilly Hassan, Youssef Haikel, Mosaad Megahed, Simeon Santourlidis, Mohamed Hassan

**Affiliations:** 1Department of Pathology, University of Mississippi Medical Center, 2500 North State Street, Jackson, MS 39216, USA; yalhmada@umc.edu (Y.A.H.); rbrodell@umc.edu (R.T.B.); 2Institut National de la Santé et de la Recherche Médicale, University of Strasbourg, 67000 Strasbourg, France; dentistenajikharouf@gmail.com (N.K.); youssef.haikel@unistra.fr (Y.H.); 3Department of Operative Dentistry and Endodontics, Dental Faculty, University of Strasbourg, 67000 Strasbourg, France; 4Department of Pharmacology and Experimental Therapeutics, LSU Health Sciences Center, New Orleans, LA 70112, USA; tflan1@lsuhsc.edu; 5College of Health Sciences, Jackson State University, 310 W Woodrow Wilson Ave Ste 300, Jackson, MS 39213, USA; alamoudi.aa89@gmail.com; 6Department of Pharmacy, Faculty of Science, Heinrich-Heine University Duesseldorf, 40225 Dusseldorf, Germany; sofie00@gmx.de; 7Department of Urology, Tulane University School of Medicine, New Orleans, LA 70112, USA; hshalaby@tulane.edu; 8Department of Chemistry, Faculty of Science, Heinrich-Heine University Duesseldorf, 40225 Dusseldorf, Germany; slh03122001@gmail.com; 9Pôle de Médecine et Chirurgie Bucco-Dentaire, Hôpital Civil, Hôpitaux Universitaire de Strasbourg, 67000 Strasbourg, France; 10Clinic of Dermatology, University Hospital of Aachen, 52074 Aachen, Germany; mmegahed@ukaachen.de; 11Epigenetics Core Laboratory, Medical Faculty, Institute of Transplantation Diagnostics and Cell Therapeutics, Heinrich Heine University Düsseldorf, 40225 Dusseldorf, Germany; simeon.santourlidis@med.uni-duesseldorf.de; 12Research Laboratory of Surgery-Oncology, Department of Surgery, Tulane University School of Medicine, New Orleans, LA 70112, USA

**Keywords:** melanoma, CSCs, PI3K, MAPK, BRAF

## Abstract

**Simple Summary:**

Melanoma is the third most common type of skin cancer. Melanoma is a heterogeneous tumor, composed of genetically divergent subpopulations. These subpopulations exist in the form of cancer stem-like cells (CSCs) and many non-cancer stem cells (non-CSCs). CSCs are characterized by their unique surface proteins and aberrant signaling pathways. The unique characteristics of CSCs are responsible for the promotion of melanoma progression, drug resistance, and recurrence. Beyond unique characteristics of CSCs, melanomas also harbor significant alterations in functional genes (BRAF, CDKN2A, NRAS, TP53, and NF1). Of these, the most common are the BRAF and NRAS oncogenes, with 50% of melanomas demonstrating the BRAF mutation (BRAF^V600E^). Although the successful targeting of BRAF^V600E^ does improve overall survival, the long-term efficacy of available therapeutic options is limited, and the development of acquired resistance is mostly common.

**Abstract:**

Melanoma is the third most common type of skin cancer, characterized by its heterogeneity and propensity to metastasize to distant organs. Melanoma is a heterogeneous tumor, composed of genetically divergent subpopulations, including a small fraction of melanoma-initiating cancer stem-like cells (CSCs) and many non-cancer stem cells (non-CSCs). CSCs are characterized by their unique surface proteins associated with aberrant signaling pathways with a causal or consequential relationship with tumor progression, drug resistance, and recurrence. Melanomas also harbor significant alterations in functional genes (BRAF, CDKN2A, NRAS, TP53, and NF1). Of these, the most common are the BRAF and NRAS oncogenes, with 50% of melanomas demonstrating the BRAF mutation (BRAF^V600E^). While the successful targeting of BRAF^V600E^ does improve overall survival, the long-term efficacy of available therapeutic options is limited due to adverse side effects and reduced clinical efficacy. Additionally, drug resistance develops rapidly via mechanisms involving fast feedback re-activation of MAPK signaling pathways. This article updates information relevant to the mechanisms of melanoma progression and resistance and particularly the mechanistic role of CSCs in melanoma progression, drug resistance, and recurrence.

## 1. Introduction

Despite improved treatment options, the prognosis for patients with advanced malignant melanoma remains poor, as measured by progression-free and overall survival [1]. Traditional therapeutics primarily target rapidly proliferating tumor cells, leaving tumor-initiating cells intact [2,3]. Consequently, tumor-initiating cells stimulate the production of new tumor clones with the propensity to disseminate to distant organs and to confer resistance to anticancer agents [4,5].

Aggressive tumor cells, including melanoma, share many characteristics with embryonic progenitors, which contribute to the mystery of tumor cell plasticity. While the multi-linage differentiation of embryonic stem cells (ESCs) is mainly controlled by a distinct microenvironment milieu leading the specification of the pluripotent ESCs [6,7], the differentiation in the case of the CSC concept refers to the ability of tumor cells to give rise to phenotypically diverse populations that reflect the histological features of the initial tumor in vivo [6]. In both embryonic and adult stem cells, differentiation is controlled by epigenetic mechanisms, and the plasticity of differentiation in these cells is associated with transcription accessibility for genes expressed in different normal tissues [7]. Abnormalities in genetic and/or epigenetic controls can lead to the development of cancer, which can be maintained by self-renewing CSCs [8,9]. Like normal stem cells, CSCs can show plasticity for differentiation [10]. CSC plasticity is mostly associated with transcription accessibility for genes that are normally expressed in different tissues, including tissues other than those from which the cancers originated [11].

In recent years, there has been an increased focus on cancer stem-like cells (CSCs) in experimental models of tumor initiation, progression, recurrence, and treatment resistance [9,12,13]. The working hypothesis suggests that CSCs, representing a small fraction of tumor cells, potentiate neoplastic clones [3]. Like other CSCs, melanoma stem-like cells (MSCs) are characterized by the expression of stemness properties-dependent protein markers and well-defined aberrant signaling pathways. 

Most available therapeutics fail to target MSCs. In fact, MSCs are genetically evolved to evade drug toxicity and to promote tumor progression and metastases [14,15]. This article focuses on the molecular mechanisms of melanoma progression and treatment resistance, and particularly the mechanistic role of CSCs in melanoma progression, drug resistance, and recurrence.

## 2. Melanoma Heterogeneity and Plasticity

Although phenotypic diversity and plasticity in melanoma have been described >40 years ago [16,17], the molecular characterization of specific phenotypic states was first determined after the functional characterization of the gene encoding MITF [18]. Consequently, it is possible to investigate the specific phenotypic states evoked by microenvironmental signals.

In addition to its significant role in the regulation of pigment cell development [19], MITF is widely discussed as a key regulator of genes leading to the regulation of melanogenesis and primary differentiation-associated function of melanocytes [20,21]. Also, accumulated evidence indicated that deregulation of either MITF expression or activity can cause melanocyte dedifferentiation [22,23]. Beyond its role in melanoma and melanocytes differentiation, MITF has been reported to be essential for the regulation of genes implicated in several biological processes such as survival [24], cell cycle control [25], invasion [26], autophagy [27], senescence bypass [28], and DNA damage repair and chromosome stability [28,29].

Tumor heterogeneity is widely documented to play an imperative role in cancer development, evolution, and resistance to therapy. As one of the most heterogeneous human cancers, melanoma demonstrates high levels of biological complexity during disease progression. As result, melanoma cells undergo genetic, epigenetic, and/or phenotypic modification to survive in the human body. In addition to the aforementioned melanoma alterations, the microenvironment of melanoma cells plays a crucial role in the regulation of melanoma initiation, progression, treatment resistance, and recurrence [30,31]. 

The study of single-cell genotyping demonstrated a complex clonal diversity among tumor cells, a phenomenon that is recognized as tumor heterogeneity; “the same tumor cells exhibit different morphological and phenotypic profiles” [32,33]. Tumor heterogeneity is a tumor phenomenon that refers to the existence of subpopulations of cells with different genotypes and phenotypes that can exhibit different biological behaviors within a primary tumor and its metastases or between tumors of the same histopathological subtype in the form of intra- and inter-tumor phenotypes [34,35,36].

Melanoma has the highest mutation frequency among human cancers, which contributes to the development of significant melanoma heterogeneity [37,38,39,40,41,42]. 

Intratumoral heterogeneity refers to development of tumor subpopulations with variable genetic traits in the same tumor, intertumoral heterogeneity refers to the differences between lesions in the same patients [32,33,37,43,44], and phenotypic heterogeneity results from irreversible changes in tumor cells within a homogenous population in response to microenvironmental signals without undergoing genetic alterations [45,46,47]. 

Genetic intratumoral heterogeneity results from genomic instability in the form of frequent mutation of genes encoding for key proteins of the aberrant signaling pathways linked to the development of genetically divergent subpopulations [48,49]. The most common examples of the genetic intratumoral heterogeneity are the melanoma subpopulations bearing mutant (Mut) protein and its wild-type (WT) counterpart such as BRAF^Mut^ or BRAF^WT^ [50,51,52,53], KIT^L576P^, KIT^WT^ [54], BRAF^V600E^/NRAS^WT^ or BRAF^WT^/NRAS^Q61R^ [48], NRAS^G13R^ or NRAS^WT^ [49], as well as melanoma subpopulations with heterogenous expression of BRAF^V600E^ [53].

Epigenetic intertumoral heterogeneity refers to tumor subpopulations with epigenetic variation [32]. The most reported examples for epigenetic intratumoral heterogeneity are those describing melanoma subpopulations bearing RASSF1A, CDKN2A, DAPK, MGMT, and RB1 genes with hypermethylated promoters [55,56] or melanoma subpopulations with heterogeneous expression of melanoma-associated antigen A3 (MAGE-A3) as a consequence of differential methylation of the MAGE-A3 promoter [56].

The microphthalmia-associated transcription factor MITF has been widely reported to be the master regulator of melanocyte biology in addition to being one of the key factors that is essential for the regulation of melanoma progression and invasion [32]. Melanoma is one of the most genetically and phenotypically heterogeneous cancers at inter-patient, inter-tumor, and intra-tumor level [57,58]. In addition to its role as key regulator of melanoma progression and invasion, MITF is one of the main determinants of melanoma heterogeneity and therapy resistance [57,58,59]. Consequently, the genetic, epigenetic, and phenotypic heterogeneity of melanoma result from the development of melanoma subpopulations within a tumor and can display remarkable variability in their phenotypic traits [60]. These heterogenous melanoma subpopulations are characterized by the expression of MITF^high^ and MITF^low^ proteins [61,62,63], MITF and BRN2 (non-canonical melanoma tumor-suppressor) proteins [64,65,66], and MITF and PAX3 [65] proteins. Also, phenotypic intertumoral heterogeneity includes subpopulations that can undergo phenotypic transition from one subpopulation to another. This includes the transition from MITF^high^/NF-κB^low^ to MITF^low^/NF-κB^high^/AXL^high^ during the development of melanoma resistance [67,68], transition from primary melanoma that expresses ZEB2^high^/SNAIL2^high^/ZEB1^low^/TWIST1^low^ to metastatic melanoma expressing ZEB2^low^/SNAIL2^low^/ZEB1^high^/TWIST1^high^ [69], transition of melanoma of the ABCB5^+^ subpopulation to the melanoma ABCB5^−^subpopulation [70,71,72,73], and transition of the melanoma CD133^+^ subpopulation to the melanoma CD133^−^ subpopulation [73,74,75,76,77]. Accordingly, the development of phenotypic intratumoral heterogeneity, in response to treatment with BRAF inhibitors, is common. The transition of the melanoma MART-1^neg^/NGFR^high^ subpopulation to the MART-1^neg^/NGFR^neg^ subpopulation [78,79], or the transition of from NRAS^WT^/BRAF^V600E^ to NRAS^G13R^/BRAF^V600E^ [80] following treatment with BRAF inhibitors, has also been reported.

Cancer cells can exhibit a high level of plasticity or the ability to dynamically switch between CSC and non-CSC states and even among different subsets of CSCs. Plasticity gives melanoma cells the ability to dynamically switch between a differentiated state with limited tumorigenic potential and an undifferentiated or cancer stem-like cell state (CSC) that is responsible for long-term tumor growth [81]. In addition to the ability to transit into distinct CSC states with different competence to evade drug toxicity to disseminate to distant organs, cancer cell plasticity has been shown to be linked to the epithelial-to-mesenchymal transition-like program in melanoma that relies not only on cell-autonomous mechanisms but also on signals provided by the tumor microenvironment and/or signals induced in response to therapy [82,83] in response to active mutations in key molecules of both MAPK and PI3K/AKT/PTEN pathways [84,85].

The development of cell cycle heterogeneity and the enhancement of cell differentiation and metabolic reprogramming contribute to the evolution of the phenotypic drug resistance of melanoma [85,86,87]. Figure 1 describes the mechanisms of cellular plasticity that allows for the adaptation of melanoma cells to a variety of environmental stresses.

## 3. Mechanisms of Tumor Progression and Drug Resistance

Drug resistance is one of the largest challenges to melanoma treatment. Melanoma initiation and progression is mediated via genetic and epigenetic alterations to the key molecules in multiple signaling pathways such as RAS/RAF/MAPK, JNK, PI3K/Akt, and Jak/STAT pathways [86,87]. Likewise, dysregulation of MITF protein results in the development of melanoma progression and drug resistance [88].

The analysis of melanoma circulated tumor DNA (ctDNA) using next-generation sequencing (NGS) has been used as a reliable tool to monitor a driver mutation as a predictor marker for survival high-risk stage III cutaneous melanoma patients [89,90] and drug resistance [91]. The most mutated genes, which are associated with the development of resistance to targeted therapy in melanoma include CDKN2A [92,93], RB1 [92,94], PIK3CA, AKT3, HOXD8 [95], PAX5 [93], MAP3K8 [96], and MITF [95]. All these genes are either involved in the regulation of MAPK and PI3K/AKT signaling pathways in addition to serving as tumor suppressors affecting drug resistance in other cancers including Homeobox protein Hox-D8 (HOXD8) [97] and Paired Box 5 (PAX5) [98].

The most common genetic alterations result from frequent mutations to the DNA sequence of significant genes, while epigenetic alterations are mediated by cytosine methylation of DNA regulatory regions [99,100].

Genetic alterations involve inherited mutation to melanoma development, particularly familial melanomas [101]. These genetic alterations result from familial/inherited and somatic CDKN2A mutations to both p14^ARF^ and p16^INK4A^, which impact melanoma suppression [102,103]. Although the CDKN2A and CDK4 genes have been primarily linked to familial melanoma, the contribution of these genes only accounts for a small percentage of familial melanoma [101,104]. The main function of p14^ARF^ is to restrict cell proliferation via a p53 stabilization-dependent mechanism, which induces the cyclin-dependent kinase inhibitor p21 [105,106]. The main function of p16^INK4A^ is to control cell proliferation by inhibiting cyclin-dependent kinases 4 and 6 (CDK4/6) and cyclin D1 [107,108]. Accordingly, CDKN2A mutations are common in melanoma and even reported in 8 to 57% of familial melanoma cases [109,110]. Somatic mutations to key genes such as BRAF are common risk factors associated with melanoma development in more than 5% of patients with BRAF mutation [111,112]. The frequent mutation of BRAF, particularly of BRAF^V600E^, has been reported in benign nevi, the precursors for melanoma genesis [113,114].

Epigenetic-dependent mechanisms such as methylation, chromatin modification, and remodeling are essential for the regulation of melanoma progression and drug resistance [115,116].

In addition to histone modification, noncoding RNA (ncRNA) expression, chromatin remodeling, and nucleosome positioning, most epigenetic changes are mediated by aberrant methylation of DNA-dependent mechanisms via the addition of a methyl group to the fifth carbon position of a cytosine molecule, which leads to generation of 5-methylcytosine (5-mC) molecules that constitute approximately 2–8% of the total cytosines of DNA regulatory sequences of interest [117,118].

Analysis of DNA methylation in melanoma cell lines has revealed a large group of hypermethylated genes, one of which includes the MITF gene [18,119]. MITF is significantly involved in the regulation of multiple biological processes, including melanoma differentiation, proliferation, migration, and senescence [120,121,122]. Likewise, the hypermethylation of the phosphatase and tensin (PTEN) homologue promoter plays a key role in the regulation of melanoma progression and resistance [123,124]. PTEN is the suppressor of PI3K, whose loss is associated with PI3K activation that, in turn, plays an essential role in the development of non-inherited melanomas [125,126].

The resistance of any tumor results from the development of primary and/or acquired resistance mechanisms. As mentioned, the occurrence of primary resistance in melanoma is attributed to the accumulation of both genetic and epigenetic alterations to tumor cells and significant changes in their microenvironment [127,128], while the development of acquired/adaptive resistance results from tumor treatment-induced genetic and epigenetic alterations to key molecules of aberrant signaling pathways in tumor cells [129,130].

Primary resistance development is mediated in great part by tyrosine-kinase-dependent phosphorylation of tyrosine residues that drive various cellular functions, including proliferation, differentiation, migration, and survival. Activation of tyrosine kinases is mediated by receptor tyrosine kinase (RTK) and/or non-receptor tyrosine kinase (NRTK)-dependent mechanisms [131,132,133,134,135]. RTK activation is strictly regulated and well balanced via ligand stimulation, chromosomal rearrangement, point mutations and amplification to RTK/NRTK genes [131,132,133,134,135,136]. RTK activation is mediated by variable mechanisms and via a cascade of phosphorylation events leading to the enhancement of cell growth, migration, differentiation, survival, or apoptosis. The most common RTK activation mechanisms include ligand-stabilized dimerization or oligomerization of their protein monomers [135,136]. Dimerization-dependent activation of RTK is mediated through the phosphorylation of tyrosine residues located in the kinase activation loop or juxta membrane domain of transmembrane receptors [135,136]. Phosphorylation of the tyrosine residues of RTKs is essential to trigger the activation of Ras/Raf/MEK/ERK and PI3K/AKT pathways [131,137].

Acquired resistance for any tumor type occurs via two different biological processes; one is early intrinsic/adaptive tumor resistance, while the other is known as late acquired resistance. Intrinsic resistance of tumors like melanoma results from the re-activation of RTK-dependent pathways (i.e., Ras/Raf/MEK/ERK and PI3K/AKT) as consequence of treatment with their specific inhibitors [138,139,140]. While a proportion of patients are intrinsically resistant to BRAF inhibitors, most patients who initially respond to BRAF inhibitors exhibit acquired resistance once treatment is initiated [139,140]. The development of BRAF inhibitor-associated acquired resistance is mediated by variable mechanisms. These mechanisms are mostly mediated by either upstream or downstream signaling, leading to re-activation of the MAPK pathway via BRAF-dependent and -independent mechanisms [141,142]. The upstream signaling-dependent mechanisms are mediated by the activation of ARAF and CRAF kinases to replace the inhibited BRAF^V600E^ functioning [143,144,145], while downstream signaling-dependent mechanisms are mediated by ERK negative feedback effects on RAS that restore RAS activity and enhance the formation of BRAF^V600E^ dimers. Of note, BRAF inhibitors can only bind one component of each dimer, a mechanism which allows the unbound BRAF monomers to interact with CRAF monomers to form BRAF-CRAF heterodimers that ultimately trigger the re-activation of ERK signaling to reduce long-term BRAF inhibitor efficacy [146,147].

In addition to the above-mentioned mechanisms, other mechanisms such as the amplification of BRAF mutations [148] and/or alternative splicing of BRAF gene, are involved in the re-activation of the MAPK pathway, leading to the development of melanoma acquired resistance to BRAF inhibitors [149]. Also, both insulin growth factor-1 receptor (IGF-1R) and PI3K/AKT pathways have been reported to be involved in the development of melanoma resistance [150,151,152]. The possible mechanisms of melanoma progression and drug resistance are summarized in Figure 2.

## 4. Cancer Stem Cells

Tumors are unique and complex ecosystems in which heterogeneous cell subpopulations with variable molecular profiles, aggressiveness, and proliferation potential coexist and interact in addition to exhibiting some self-renewal properties [153,154]. Two major models have been proposed to explain how tumors grow and progress [155,156]. In one of these models, namely the stochastic model, all the tumor cells are similar in their biological features, but their fates are determined by their intrinsic signals and their microenvironment-dependent signals. Although not all cancer cell progeny has the potential to behave like a cancer stem cell, they have the potential to retain plasticity to go from a non-stem cell to a stem cell-like precursor [156]. While in the hierarchical model, the cancer stem cells are biologically different and can self-renew in addition to giving rise to various progeny cells including those lacking the ability to self-renew [157]. However, the hierarchical model is often considered to be the most common model for sustained tumor propagation rather than the stochastic model [156].

Like normal stem cells, CSCs are hierarchically organized at the cellular level in origin tissues as a small fraction of genetically divergent subpopulations [158,159]. The development of these subpopulations results from the segregation of genetic material of functional genes to intrinsically asymmetric cell division of stem cell lineage to produce two daughter cells, both different in their genetic materials and phenotype [160,161]. The process of asymmetric cell division is genetically programed to produce one cell with stemness properties that is recognized as a CSC and one cell that is recognized as a non-CSC [124]. CSCs are adopted and genetically programed to grow continuously and divide indefinitely, whereas non-CSCs are characterized by their limited cell division [160,161]. While non-CSCs are sensitive to anti-cancer agents, CSCs evade drug toxicity and relocate and metastasize to distant organs [4,14,74].

Many potential biomarkers of CSCs have been identified based on their expression in human solid tumors. Potential CSC markers include neural crest nerve growth factor/neurotrophin receptor CD271 [4,78,79,162]. CD271^+^ melanoma cells are characterized by their tumorigenicity [4,78,79], and propensity to metastasize to the brain [130]. CD20, the cell surface marker of normal B cells, exhibits elevated expression in melanoma [4,14,163,164,165]. Transcription factors such as Nanog and Oct3/4 transcription factors have been found to be markedly elevated in melanospheres when compared to adherent melanoma cells [166,167]. Likewise, the activation of signaling pathways is common in normal stem cells such as Wnt, and Notch and Hedgehog are also activated in melanoma CSCs [168,169,170,171]. Aldehyde dehydrogenase (ALDH1) has also been identified as a potential marker of CSCs associated with multidrug and/or immunological resistance [170]. Accordingly, genes such as ABCB1, ABCB5, and ABCG2 undergo differential expression in both melanocytes and melanoma cells [4,71,171,172,173]. Finally, Sox10 expression maintains the growth of MSCs to grow as non-adherent tumorigenic spheres [174,175].

The involvement of the stem cell marker CD133 protein in the maintenance of melanoma stemness properties and drug resistance is mediated by its C-terminal domain, which contains tyrosine binding sites located on tyrosine 828 (Tyr828) and tyrosine 852 (Tyr852) residues [74,176]. These two tyrosine residues are phosphorylation targets of the non-receptor tyrosine kinase (NRTK) Fyn [176]. The contribution of CD133 to the regulation of CSC functions such as self-renewal, differentiation, and drug resistance are likely mediated by the NRTK, Fyn-dependent mechanism via the phosphorylation of Tyr828 residue located on the cytoplasmic domain of CD133 [74,176]. Our laboratory has demonstrated that the phosphorylation of Tyr828 is essential to trigger the activation of PI3K and its downstream-dependent signaling pathways in melanoma [74]. Many studies have demonstrated that increased CD133 expression is associated with high tumorigenicity and metastatic potential for melanoma cells [75,177,178,179]. Also, CD133 protein has been implicated in the regulation of tumor resistance [180,181,182,183].

CD133-expressing CSCs have been shown to exhibit resistance to chemotherapy and radiation therapy in addition to being associated with poor prognosis in various cancers [182]. We and others demonstrated that CD133+ cancer cells confer resistance to many chemotherapeutic agents such as caffeic acid phenethyl ester [4], taxol [14], and fotemustine [74]. Accordingly, CD133-dependent mechanisms have been shown to be involved in the development of melanoma resistance to chemotherapy [74].

Thus, understanding the mechanisms which are involved in the regulation of CSC growth and maintenance may help develop innovative therapeutic approaches for melanoma treatment.

The preliminary prevailing hypothesis suggests that tumor-initiating cells are derived from normal stem cells rather than from progenitor cells; the development of CSCs from normal stem cells and from progenitor stem cells has been shown [156,184,185]. Figure 3 outlines the pathways by which MSCs develop from either normal stem cells or cancer progenitor cells. In addition to their origin and unique properties, CSCs serve as an experimental model by which the mechanisms of tumor progression, recurrence, and treatment resistance can be investigated.

## 5. Mechanisms of Melanoma Treatment Failure and Recurrence

Mechanistically, the role of CSCs in tumor initiation, metastasis, and therapy resistance have been demonstrated [4,14,74,174]. Stemness property-associated proteins like CD133 and CD271 have been purposed to play a functional role in melanoma progression, metastasis, and drug resistance [74,75,79,165,186]. Of note, approximately 50% of melanoma cases involve BRAF mutations [187]. BRAF inhibitors (e.g., vemurafenib, dabrafenib) have become the preferred therapy for melanoma patients with mutated BRAF^V600E^ [188,189,190,191,192]. Unfortunately, while the successful targeting of BRAF^V600E^ improves overall survival [188,193], acquired resistance development is common in most patients. This type of resistance is the direct consequence of secondary NRAS mutations [194,195] in response to BRAF inhibitor induced-ERK negative feedback effects on RAS as well as RTK-dependent RAS activation, which triggers the activation of both ARAF and CRAF, BRAF amplification, and activation of pathways like PI3K [196,197,198].

Melanoma progression, metastasis, treatment failure, and recurrence are attributed to the presence of genetically divergent MSC subpopulations with stemness properties [199,200]. First assessment of melanoma patients treated with BRAF inhibitors revealed that 20% targeted the BRAF^V600E^ mutation [201,202]. Thus, the development of intrinsic resistance in melanoma patients with the BRAF^V600E^ mutation results from the presence of a portion of cells (e.g., tumor-initiating cells/CSCs) that confer drug resistance [201,203].

While treatment with BRAF inhibitors displays initial tumor regression in most melanoma patients, complete tumor regression rarely occurs [204,205]. The main cause of this noted poor prognosis is attributed to BRAF inhibitor-mediated compensatory mechanisms. These compensatory mechanisms include elevation of ARAF and CRAF proteins, a mechanism through which melanoma cells sustain the activation of MAPK activity during BRAF inhibitors treatment [206,207,208]. Although BRAF inhibitor-resistant cells remain sensitive to MEK inhibitors, the resistance of these melanoma cells to various structurally different MEK inhibitors can be mediated by the generation of the compensatory mechanism via activation of RTKs such as IGF-1R [205,206,207,208]. Like other cells with a melanocyte origin, melanoma cells express IGF-1R. Importantly, MEK inhibitor-resistant melanoma cells demonstrate higher surface levels of IGF-1R when compared to BRAF-sensitive melanoma cells [205,206,207,208]. As such, IGF-1R is known to play an important role in tumor progression and drug resistance [209]. IGF-1R-induced effects are mediated via activation of MAPK and PI3K signaling [206,207,208,209]. This crosstalk between the MAPK and IGF-1R/PI3K/AKT signaling pathways promotes the survival and expansion of BRAF-inhibitor-resistant cells.

The most widely recognized challenge of current therapies for solid tumors involves this development of drug resistance and tumor recurrence. Common molecular mechanisms involved in therapeutic resistance include increased drug metabolism and drug efflux; enhanced repair capacity of damaged DNA; re-activation of drug targets; overactivation of growth and survival signaling pathways; amplification of genetic mutations; and impaired activity of apoptosis/autophagy-dependent pathways [142,152]. Tumor heterogeneity is also an important driver of drug resistance [32,80,210,211]. In contrast to non-CSCs, CSCs are characterized by specific features that allow them to escape drug toxicity [211,212]. These specific features include expression of different members of the ATP-binding cassette (ABC) transporters such as ABCB5, ABCG1, ABCG2, and ABCG5, all “efflux pumps”, to extrude drugs from cells into the microenvironment [213,214]. In addition to the ABC transporters, high levels of ALDH activity modulate CSC resistance to anti-cancer agents [160,215,216]. CSCs have been shown to be resistant to radiation therapy, based on their ability to repair damaged DNA [74,217,218]. Finally, inhibition of fotemustine-induced DNA damage in CD133^+^ melanoma subpopulations is associated with the activation of checkpoint kinases CHK1 and CHK2 [74]. Taken together, these observations strongly support the notation that melanoma CSCs represent a target for melanoma treatment with standard therapies being combined to be directed against the tumor bulk.

Current approaches to cancer immunotherapy include the non-specific stimulation of antitumor immune response through stimulation of endogenous effector cells via cytokine-dependent mechanisms, active immunization, adoptive immunotherapy, and the targeting of immune checkpoints or immune regulatory molecules [219]. The most common cytokine-based therapy involves targeting interleukin-2 (IL-2), which was the first type of immunotherapy approved for melanoma treatment [220,221,222]. Various systemic immune therapies have also been examined for patients with stage II or III melanoma, particularly for patients with a high risk of systemic recurrence [223]. The most common immunotherapeutic of this type involves targeting interferon α-2b and peginterferon α-2b [223,224].

Accordingly, the combination of chemotherapy with either immunotherapy or biotherapy in metastatic melanoma treatment has been explored in multiple trials; notable examples include the combination of cisplatin, vinblastine, and dacarbazine with IL-2 and interferon-α-2b immunotherapy [225,226,227,228].

Tumor vaccination-based therapies for melanoma treatment include multiple approaches. Unfortunately, most protein- or peptide-based vaccines lack significant immunogenicity and are unable to induce a robust immune response in monotherapy [229,230]. Likewise, the administration of customized vaccines, derived from tumor patient whole-cell lysates, does not show any clinical benefit in randomized controlled trials [231]. Other approaches include recombinant vectors, which encode entire genes or the antigenic epitope [232], and active vaccination of dendritic cells pulsed with tumor cell RNA, DNA, or cell lysate [233,234].

The treatment of melanoma with adoptive cell therapy has also been employed. Adoptive cell therapy involves the clinical utilization of autologous engineered T cells expressing receptors that specifically recognize various tumor-associated antigens such as melanocytic protein Melan-A/MART-1-TCG genes or that secrete specific cytokines [232,233,234,235].

Immune checkpoint modulators and immune modulator molecules have also been investigated [234]. Immune checkpoints target cell surface proteins such as cytotoxic T-lymphocyte antigen-4 (CTLA4) and programmed cell death-1 (PD-1) [235,236,237,238]. These proteins are implicated in the regulation of immune response initiation and duration [194]. Inhibition of these immune checkpoint inhibitors by anti-CTLA4 and anti-PD-1 antibodies demonstrated significant tumor regression and long-term durable cancer control when compared to other systematic therapies [238,239].

The expression of CSC makers in human tumor tissues is closely associated with the number of tumor-infiltrating immune cells [240,241]. As such, the CSC immune response is frequently compromised by immune evasion properties. These immune evasion properties are mediated by CSC-secreted immunosuppressive factors as well as the ability of CSCs to recruit immunosuppressive noncancerous cells [242,243,244], enhancing immune evasion strategies.

CSC-mediated immunosuppression both in vitro and in vivo allows CSCs to evade anti-tumor immune-mediated reactions [245]. Melanoma is characterized by their expression of a variety of antigens, including differentiation and cancer testis antigens such as MelanA/Mart1, HMB45, tyrosinase, gp100, and NYESO1 [246,247]. MSCs have the potential to initiate the activation of several mechanisms to maintain tumor survival and escape the patient’s immune reactions [244,248], via their mechanistic role in the suppression of anti-tumor immune reactions. One of these mechanisms involves the expression of multidrug resistance (MDR) genes and related functional transporters. MDR is the ability to mount resistance to multiple, structurally unrelated therapeutic drugs with different mechanisms of action [249,250,251]. MDR pathways in melanoma are responsible for the decrease in intracellular drug accumulation and is mediated by energy-dependent efflux pumps via ABC transporters that translocate solutes across the cellular membrane [252,253]. We and others have shown the involvement of the ABCB5 transporter in the modulation of melanoma resistance to anti-cancer agents like caffeic acid phenethyl ester [4] and doxorubicin [4,254,255]. Accordingly, blockage of ABCB5 enhances intracellular drug accumulation in melanoma cells sensitive to the therapeutic effects of BRAF inhibitors [256].

Resistance of MSCs to BRAF and MEK inhibitor-induced cell death has been reported [257,258]. Increased activity of Stearoyl-CoA-desaturase 1 (SCD1), the rate-limiting enzyme in the formation of monounsaturated fatty acids, has been shown to overcome BRAF and MEK inhibitor-induced death of MSCs [259].

The CD133 protein remains the most important identified stem cell marker. Functional analysis of this protein suggests that CD133 is a key tumor progression and treatment-resistance-driving signaling protein in melanoma [74], via a mechanism mediated by binding of the regulatory subunit of PI3K, p85, to the Tyrosine 828 (Tyr^828^) residue located on the cytoplasmic domain of the CD133 protein [74]. In addition to its contribution to the regulation of melanoma proliferation and metastasis [260,261,262], the overexpression of CD271 is associated with the induction of stem-like quiescence [160,260,261,262]. CD271 expression is lost in early progression when melanoma cells invade the dermis [263]. The low expression of CD271 is likewise associated with phenotype switching in melanoma [261].

Both tumor autonomous mechanisms and adaptive survival signaling may be related to melanoma progression and drug resistance. Once the processes of melanoma progression have been initiated, tumor cells tend to detach from their natural binding partners, namely keratinocytes, to interact with host cells such as fibroblasts and endothelial cells [2,264], inducing the secretion of the hepatocyte growth factor (HGF), endothelial growth factor (EGF), neuregulin (NRG), and IGF-1R [265,266,267]. Secretion of these tumor-derived growth factors often occurs in response to treatment with BRAF inhibitors, a common mechanism by which melanoma cells confer resistance to BRAF inhibitors [268,269].

Accordingly, melanoma treatment with BRAF inhibitors is theorized to remodel the host environment [151,270]. Clinically utilized BRAF triggers contradictory mitogen-activated protein kinase (MAPK) signaling pathways as well as triggers differentiation in normal skin fibroblasts, particularly those bearing BRAF^WT^ cells [271]. Thus, such bidirectional signaling between tumor and host cells plays an active role in the configuration of these mechanisms, which is essential for the development of adaptive resistance to BRAF inhibitors [197,272].

Treatment with BRAF and/or MEK inhibitors mediates both autocrine and paracrine effects, which in turn trigger a stress-induced senescent phenotype in different melanoma subpopulations [273,274]. Cellular senescence is a phenomenon that occurs in melanoma patients during and/or after melanoma treatment has been initiated [275,276].

Cellular senescence is an autonomous tumor suppressor mechanism associated with the stabilization of cell cycle arrest [277,278]. Senescent cells are characterized by their ability to secrete variable factors that can change tumor cells and their microenvironment, allowing tumor cells to evade the toxicity of anti-cancer agents and subsequently grow and metastasize. Senescence-associated secretory phenotype (SASP) is associated with senescent cells wherein cells secrete high levels of inflammatory cytokines, immune modulators, growth factors, and proteases [279,280,281,282,283,284,285]. SASP may consist of exosomes and ectosomes containing enzymes, microRNA, DNA fragments, chemokines, and other bioactive factors [286]. Previous and current reports indicate that senescent cells are highly secretory cells and drive a range of different functions via SASP-dependent mechanisms [287,288]. The formation of SASP composition is dynamically and spatially regulated; thus, changing SASP composition can determine the beneficial and detrimental aspects of the senescence program, tipping the balance to either an immunosuppressive/pro-fibrotic environment or pro-inflammatory/fibrolytic state. The temporal and spatial regulation of SASP and NOTCH signaling may regulate SASP composition [289,290]. As the composition of SASP is dynamically and spatially regulated, a change in SASP can determine whether the senescence program progresses to an immunosuppressive/profibrotic environment or proinflammatory/fibrolytic state [287,288]. In summary, SASP has the potential to trigger tumor growth through the change in tumor microenvironment composition that ultimately influences treatment outcomes.

Melanoma cells with a senescent phenotype are characterized by the upregulation of two genes, one of which is matrix metalloprotease 2 (MMP2) [291,292]. Vemurafenib-induced MMP-2 has been shown to enhance melanoma invasiveness via an alteration in tumor microenvironment [280]. The other gene encodes monocyte chemoattractant protein 1 (MCP1). Functionally, MCP1 is involved in the initiation and subsequent activation of the poly (ADP-ribose) polymerase-1 (PARP-1)/NF-kB signaling cascade, which is essential for the regulation of tumor progression and dissemination to distant organs [287,291,293].

BRAF, ALK, or EGFR kinase inhibitors induce a complex network of secreted signals (secretome) in most tumor types including melanoma [294,295]. In addition to their role in the development of acquired resistance, the BRAF, ALK, or EGFR inhibitor-induced secretome results in an incomplete tumor regression and the stimulation of tumor growth and metastasis [291,292].

For example, vemurafenib-induced MMP-2 has been reported to enhance melanoma invasiveness and alter the tumor microenvironment, which triggers tumor resistance mechanisms [285,296]. Figure 4 provides a brief description of the possible mechanisms regulating the feedback activation of tumor growth factor (TGF) receptors during BRAF and MEK inhibitor-based therapy in MSCs.

## 6. Conclusions

Melanoma progression and treatment resistance are attributed to tumor heterogeneity-dependent mechanisms that are mostly associated with development of genetically divergent subpopulations. These subpopulations exist in the form of CSCs or non-CSCs. Like other CSCs of any solid tumors, MSCs are identifiable via stemness specific markers. Thus, MSCs are central to tumor development, drug resistance, and recurrence. The development of MSCs is attributed to genetic and epigenetic changes, which lead to deregulation of several signal transduction pathways, including MAP kinase and PI3K/AKT. While BRAF/MEK and PI3K/AKT pathway inhibitors represent an intriguing therapeutic option for patients with metastatic melanoma, the success of these therapeutics is reduced by the development of acquired resistance via MSC-dependent mechanisms. These resistance mechanisms result from treatment pressure-induced by either BRAF/MEK or PI3K/AKT pathway inhibition. Both PI3K/AKT/mTOR and RAF/MEK/ERK signaling cascades are derived from numerous feedback loops and are interconnected at multiple points of crosstalk. Inhibition of one of these pathways can result in the activation of the other signaling cascade. The re-activation of the MAP kinase pathway occurs either via bypass-dependent activation of MEK by c-RAF/RAF1 or amplification and relative splicing of BRAF. Conversely, activation of the PI3K/AKT pathway results from the inhibition of the MAP kinase pathway. Thus, dual targeting of both pathways may improve melanoma treatment efficacy and lead to better clinical outcomes.

## Figures and Tables

**Figure 1 cancers-16-00470-f001:**
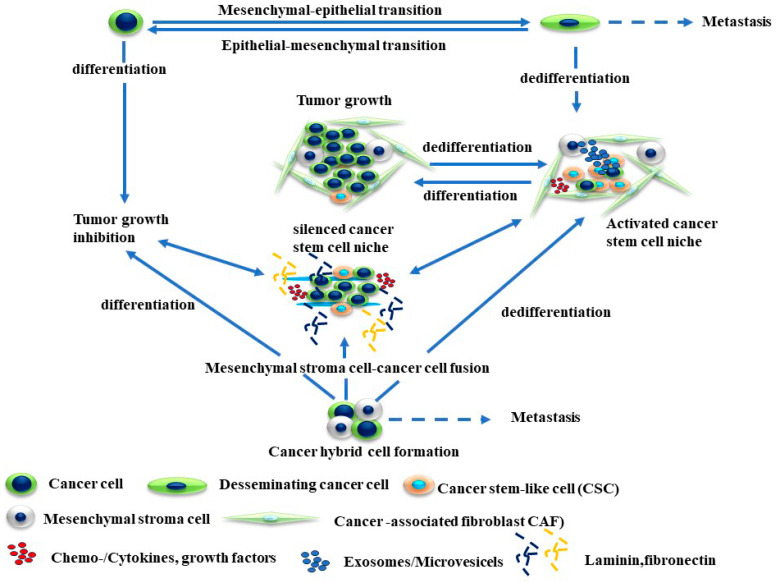
Mechanisms of melanoma plasticity. The development of melanoma plasticity is mediated by genetic and epigenetic mechanisms that can derive epithelial–mesenchymal transition (EMT), mesenchymal–epithelial transition (MET), and cancer cell fusion. Consequently, the development and accumulation of populations of cancer cell progenitors or cancer stem cells (CSCs) can form a cancer cell niche. The enrichment of CSC populations in an activated niche is the main source for their differentiation or trans- or dedifferentiation, and metastasis of melanoma cells can take place at the same time in distinct compartments of tumor cells.

**Figure 2 cancers-16-00470-f002:**
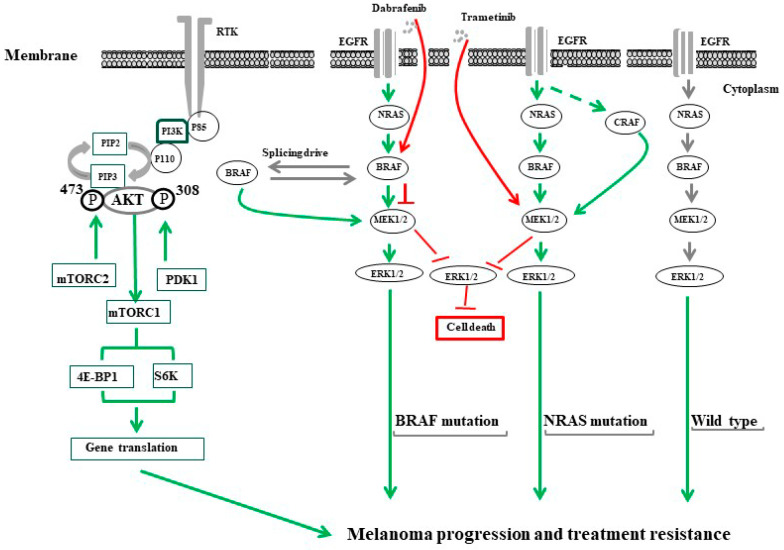
The common mechanisms of melanoma progression and treatment resistance. PI3K/AKT/mTOR and RAS/RAF/MEK/ERK pathways are the most characterized tumor growth/treatment resistance-driving signals in melanoma. The activation of PI3K/AKT/mTOR is tightly controlled via a multistep process that can be initiated through tyrosine kinase receptors (RTKs) such as IGF-1R. The stimulation of IGF-1R by its ligand triggers the activation of PI3K via their catalytic (p110) and regulatory (p85) subunits leading to the conversion of phosphatidylinositol (3,4)-bisphosphate (PIP2) lipids to phosphatidylinositol (3,4,5)-trisphosphate (PIP3) allowing PDK1 to phosphorylate the tyrosine 308 (T308) residue and mTORC2 to phosphorylate tyrosine 473 (T473) in the activation loop of AKT. Activated AKT then activates mTORC1 that, in turn, enhances the activation of the eukaryotic translation initiation factor 4E binding protein 1 (4EBP1), and ribosomal protein S6 kinase (S6K). The activation of the Ras/RAF/MEK/ERK signaling pathway is mediated by the stimulation of epidermal growth factor (EGF) by its ligand for BRAF wild type or by active mutation in the cells bearing a NRAS mutation. The treatment of melanoma cells bearing BRAF^V600E^ with specific inhibitors (e.g., dabrafenib) stimulates melanoma cells to overcome the inhibitory effect of BRAF through the generation of BRAF isoforms via alternative splicing. Treatment of patients with a NRAS mutation (NRAS^Q61R^) with MEK inhibitors (e.g., trametinib) generates ERK negative feedback effects on RAS, which creates a bypass mechanism through activation of ARAF and/or CRAF to replace functional BRAF.

**Figure 3 cancers-16-00470-f003:**
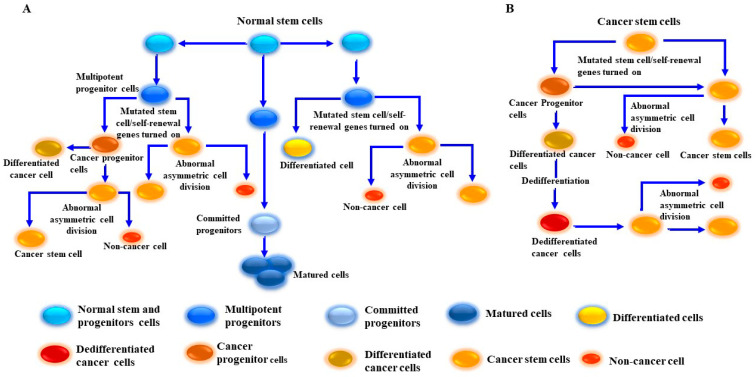
Proposed model for cancer stem-like cell (CSC)/melanoma stem cell (MSC) generation from normal and cancer cells. (**A**) Generation of CSCs/MSCs results from the transformation of normal stem/progenitor cells into undifferentiated cancer cells via multiple genetic mutations and dedifferentiation-dependent mechanisms. (**B**) Generation of MSCs from cancer cells. This model describes the generation of MSCs from tumor cells through the activation of aberrant signaling pathways via driver mutations to tumor growth signaling pathways and activation of self-renewal genes. MSCs become educated to divide into tumor progenitor cells/MSCs. The produced MSCs possess genetic properties which allow for the division into one differentiated cell and one MSC. Once the dedifferentiation process of the differentiated cell is complete, the dedifferentiated cells can be transformed into MSCs. CSCs/MSCs undergo abnormal asymmetric cell division to produce two daughter cells, one CSC and one non-CSC.

**Figure 4 cancers-16-00470-f004:**
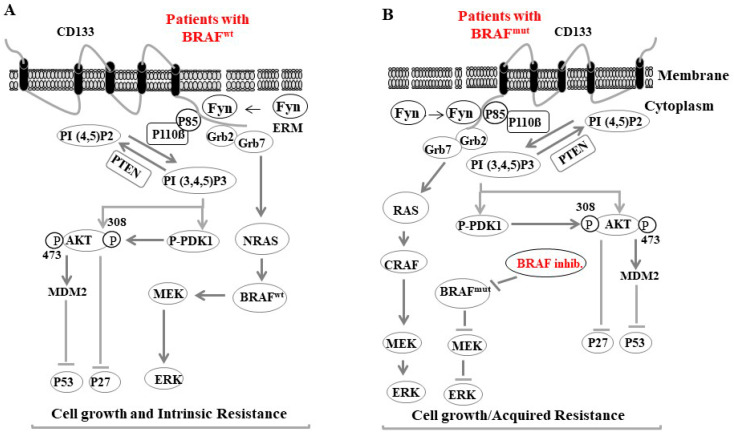
A proposed model for Fyn-CD133 signaling to the PI3K pathway induces melanoma intrinsic and acquired resistance to anti-cancer agent in melanoma cells. (**A**) Fyn stimulated CD133 signaling to the PI3K pathway in melanoma population bearing the BRAF wild type (BRAFWT) is mediated by Fyn/CD133/PI3K/PDK1/AKT-induced inhibition of p27, Fyn/CD133/PI3K/AKT/MDM2-induced ubiquitination of p53 and Fyn/CD133/Grb2/Grb7/NRAS7BRAF/MEK/ERK-induced cell growth. (**B**) Intrinsic resistance of melanoma cells: Fyn stimulated CD133 signaling to the Grb2/Grb7/NRAS7BRAF/MEK/ERK pathway is responsible for acquired resistance of BRAF inhibitors (e.g., dabrafenib).

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
