# Peer review of "Mechanisms of Melanoma Progression and Treatment Resistance: Role of Cancer Stem-like Cells"

_cancers, 2024, doi:10.3390/cancers16020470_

Round 1
Reviewer 1 Report
Comments and Suggestions for Authors
The review paper aims to summarize and discuss the mechanisms of CSCs in melanoma progression, drug resistance and recurrence. But the review article is poorly organized and lacks coherence. It is more like a reading summary instead of an academic review article.
1. First the author talked about the melanoma heterogeneity, from line 78 to 137, this part seems to have no relation to CSCs. And it is not very well written. Then the next two paragraphs talked about the plasticity briefly with no details at all.
2. The second part is about tumor progression and drug resistance, and this section also has no clear correlation between each paragraph. Although the author cited lots of genes and pathways, how are they related to melanoma CSCs?
3. The third section, cancer stem cells, also lacks detailed and deep discussion.
Author Response
Editor
Cancers
Dear Editor,
Thank you very much for the encouraging comment regarding our Manuscript ID: cancers-2784052 “ Mechanisms of Melanoma Progression and Treatment Resistance: Role of
Cancer Stem-Like Cells .”
Enclosed find please our Point-for-Point response to the valuable comment of reviewer 1.
On behalf all my coauthors
Authors’ response to comment 1
The review paper aims to summarize and discuss the mechanisms of CSCs in melanoma progression, drug resistance and recurrence. But the review article is poorly organized and lacks coherence. It is more like a reading summary instead of an academic review article.
Comment 1: First the author talked about the melanoma heterogeneity, from line 78 to 137, this part seems to have no relation to CSCs. And it is not very well written. Then the next two paragraphs talked about plasticity briefly with no details at all.
Authors’response: Thank you very much for your comment. Accordingly, we started first to extend the introduction by the addition of a paragraph to the section of introduction about the similarity between embryonic progenitors and cancer cells in their characteristic and evolution. As required, we modified the Section of Tumor heterogeneity and plasticity by adding a paragraph providing more information about tumor heterogeneity and plasticity.
See the following changes:
Introduction section
See lines:67-80; the following paragraph [Aggressive tumor cells, including melanoma share many characteristics with embryonic progenitors, which contribute to the mystery of tumor cell plasticity. While the multi-linage differentiation of embryonic stem cells (ESCs) is mainly controlled by a distinct microenvironment milieu leading the specification of the pluripotent ESCs [6, 7]]., the differentiation in the light of the CSC concept refers to the ability of tumor cells to give rise to phenotypically diverse populations that reflect the histological features of the initial tumor in vivo [6]. In both embryonic and adult stem cells, differentiation is controlled by epigenetic mechanisms, and the plasticity of differentiation in these cells is associated with transcription accessibility for genes expressed in different normal tissues [7]. Abnormalities in genetic and/or epigenetic controls can lead to the development of cancer, which can be maintained by self-renewing CSCs [8, 9]. Like normal stem cells, CSCs can show plasticity for differentiation [10]. CSCs plasticity is mostly associated with transcription accessibility for genes that are normally ex-pressed in different tissues, including tissues other than those from which the cancers originated [11].] has been added to the introduction section.
References section
The following references: See lines: 643-654)
(6) Reya, T.; Morrison, S. J.; Clarke, M. F.; Weissman, I. L. Stem cells, cancer, and cancer stem cells. Nature 2001, 414 (6859), 105-111. DOI: 10.1038/35102167.
(7) Lotem, J.; Sachs, L. Epigenetics and the plasticity of differentiation in normal and cancer stem cells. Oncogene 2006, 25 (59), 7663-7672. DOI: 10.1038/sj.onc.1209816.
(8) Toh, T. B.; Lim, J. J.; Chow, E. K. Epigenetics in cancer stem cells. Mol Cancer 2017, 16 (1), 29. DOI: 10.1186/s12943-017-0596-9.
(9) Schatton, T.; Frank, M. H. Cancer stem cells and human malignant melanoma. Pigment Cell Melanoma Res 2008, 21 (1), 39-55. DOI: 10.1111/j.1755-148X.2007.00427.x.
(10) Warrier, N. M.; Kelkar, N.; Johnson, C. T.; Govindarajan, T.; Prabhu, V.; Kumar, P. Understanding cancer stem cells and plasticity: Towards better therapeutics. Eur J Cell Biol 2023, 102 (2), 151321. DOI: 10.1016/j.ejcb.2023.151321.
(11) Tang, D. G. Understanding cancer stem cell heterogeneity and plasticity. Cell Res 2012, 22 (3), 457-472. DOI: 10.1038/cr.2012.13.] have been added to the section of references.
Section of Tumor heterogeneity and plasticity
The following changes have been made:
See Lines: 94-107; the following paragraph [Although phenotypic diversity and plasticity in melanoma has been described >40 years ago [16, 17], the molecular characterization of specific phenotypic states was first determined after the functional characterization of the gene encoding MITF [18]. Consequently, it becomes possible to investigate the specific phenotypic states evoked by microenvironmental signals.
In addition to its significant role in the regulation of pigment cells development [19], MITF is widely discussed as a key regulator of genes leading to the regulation of melanogenesis, primary differentiation-associated function of melanocytes [20, 21]. Al-so, accumulated evidence indicated that deregulation of either MITF expression or activity can cause melanocyte dedifferentiation [22, 23]. Beyond its role in melanoma and melanocytes differentiation, MITF has been reported to be essential for the regulation of genes implicated in several biological processes such as survival [24], cell cycle control [25], invasion [26], autophagy [27], senescence bypass [28], and DNA damage re-pair and chromosome stability [28, 29].] has been added to the section of tumor heterogeneity and plasticity.
References section
See lines:665-699, The following references have been added:
(16) Rambow, F.; Marine, J. C.; Goding, C. R. Melanoma plasticity and phenotypic diversity: therapeutic barriers and oppor-tunities. Genes Dev 2019, 33 (19-20), 1295-1318. DOI: 10.1101/gad.329771.119.
(17) Diazzi, S.; Tartare-Deckert, S.; Deckert, M. The mechanical phenotypic plasticity of melanoma cell: an emerging driv-er of therapy cross-resistance. Oncogenesis 2023, 12 (1), 7. DOI: 10.1038/s41389-023-00452-8
(18) Hartman, M. L.; Czyz, M. MITF in melanoma: mechanisms behind its expression and activity. Cell Mol Life Sci 2015, 72 (7), 1249-1260. DOI: 10.1007/s00018-014-1791-0.].
(19) Hsiao, J. J.; Fisher, D. E. The roles of microphthalmia-associated transcription factor and pigmentation in melano-ma. Arch Biochem Biophys 2014, 563, 28-34. DOI: 10.1016/j.abb.2014.07.019.;
(20) Gelmi, M. C.; Houtzagers, L. E.; Strub, T.; Krossa, I.; Jager, M. J. MITF in Normal Melanocytes, Cutaneous and Uveal Melanoma: A Delicate Balance. Int J Mol Sci 2022, 23 (11). DOI: 10.3390/ijms23116001.;
(21) Chauhan, J. S.; Hölzel, M.; Lambert, J. P.; Buffa, F. M.; Goding, C. R. The MITF regulatory network in melano-ma. Pigment Cell Melanoma Res 2022, 35 (5), 517-533. DOI: 10.1111/pcmr.13
(22) Fernández-Barral, A.; Orgaz, J. L.; Baquero, P.; Ali, Z.; Moreno, A.; Tiana, M.; Gómez, V.; Riveiro-Falkenbach, E.; Ca-ñadas, C.; Zazo, S.; et al. Regulatory and functional connection of microphthalmia-associated transcription factor and anti-metastatic pigment epithelium derived factor in melanoma. Neoplasia 2014, 16 (6), 529-542. DOI: 10.1016/j.neo.2014.06.00.
(23) Selzer, E.; Wacheck, V.; Lucas, T.; Heere-Ress, E.; Wu, M.; Weilbaecher, K. N.; Schlegel, W.; Valent, P.; Wrba, F.; Pehamberger, H.; et al. The melanocyte-specific isoform of the microphthalmia transcription factor affects the pheno-type of human melanoma. Cancer Res 2002, 62 (7), 2098-2103.
(24) Kawakami, A.; Fisher, D. E. The master role of microphthalmia-associated transcription factor in melanocyte and mel-anoma biology. Lab Invest 2017, 97 (6), 649-656. DOI: 10.1038/labinvest.2017.9.
(25) Loercher, A. E.; Tank, E. M.; Delston, R. B.; Harbour, J. W. MITF links differentiation with cell cycle arrest in melano-cytes by transcriptional activation of INK4A. J Cell Biol 2005, 168 (1), 35-40. DOI: 10.1083/jcb.200410115.
(26) Dilshat, R.; Fock, V.; Kenny, C.; Gerritsen, I.; Lasseur, R. M. J.; Travnickova, J.; Eichhoff, O. M.; Cerny, P.; Möller, K.; Sigurbjörnsdóttir, S.; et al. MITF reprograms the extracellular matrix and focal adhesion in melanoma. Elife 2021, 10. DOI: 10.7554/eLife.63093.
(27) Möller, K.; Sigurbjornsdottir, S.; Arnthorsson, A. O.; Pogenberg, V.; Dilshat, R.; Fock, V.; Brynjolfsdottir, S. H.; Bind-esboll, C.; Bessadottir, M.; Ogmundsdottir, H. M.; et al. MITF has a central role in regulating starvation-induced au-tophagy in melanoma. Sci Rep 2019, 9 (1), 1055. DOI: 10.1038/s41598-018-37522-6.
(28) Leclerc, J.; Ballotti, R.; Bertolotto, C. Pathways from senescence to melanoma: focus on MITF sumoy-lation. Oncogene 2017, 36 (48), 6659-6667. DOI: 10.1038/onc.2017.292.
(29) Binet, R.; Lambert, J. P.; Tomkova, M.; Tischfield, S.; Baggiolini, A.; Picaud, S.; Sarkar, S.; Louphrasitthiphol, P.; Dias, D.; Carreira, S.; et al. DNA damage-induced interaction between a lineage addiction oncogenic transcription factor and the MRN complex shapes a tissue-specific DNA Damage Response and cancer predisposition. bioRxiv 2023. DOI: 10.1101/2023.04.21.53781.
Comment 2: The second part is about tumor progression and drug resistance, and this section also has no clear correlation between each paragraph. Although the author cited lots of genes and pathways, how are they related to melanoma CSCs?
Authors’response: Thank you very much for your comment. In this review, we first reviewed the current indemnified mechanisms of tumor progression and drug resistance in the context of RAS/RAF/MAPK, JNK, PI3K/Akt and Jak/ STAT pathways-dependent melanoma progression and drug resistance as background for the reader regarding to the molecular mechanisms of melanoma progression and treatment resistance (See Lines:193-293]. Next, we started with the section of CSCs, followed by a section of the Mechanisms of melanoma treatment failure and recurrence in context of cancer stem cells (See lines:394-578).
Comment 3: The third section, cancer stem cells, also lacks detailed and deep discussion.
Authors’response: Thank you very much for your comment. Accordingly, we modified the section of cancer stem cells by adding of two paragraphs:
Section of cancer stem cells
First paragraph:
See lines: 315-326; the following paragraph [Tumors are unique and complex ecosystems, in which heterogeneous cell subpopulations with variable molecular profiles, aggressiveness, and proliferation potential coexist and interact in addition to exhibiting some self-renewal properties [153, 154]. Two major models have been proposed to explain how tumors grow and progress [155]. In one of these models, namely the stochastic model, all the tumor cells are similar in their biological features, but their fates are determined by their intrinsic signals and their microenvironment-dependent signals. Although all cancer cell progeny does not have the potential to behave like a cancer stem cell, they have the potential to retain plasticity to go from a non-stem cell to a stem cell-like precursor [156]. While in the hierarchical model, the cancer stem cells are biologically different, can self- renew in addition to giving rise to various progeny cells including those lacking the ability to self-renew [157]. However, the hierarchical model is often considered to be the most common model for sustained tumor propagation rather than the stochastic model [156].] has been added
References section
See lines: 1033-143; the following references:
(153) Kashyap, A.; Rapsomaniki, M. A.; Barros, V.; Fomitcheva-Khartchenko, A.; Martinelli, A. L.; Rodriguez, A. F.; Gabra-ni, M.; Rosen-Zvi, M.; Kaigala, G. Quantification of tumor heterogeneity: from data acquisition to metric generation. Trends Biotechnol 2022, 40 (6), 647-676. DOI: 10.1016/j.tibtech.2021.11.006.
(154) Shlyakhtina, Y.; Moran, K. L.; Portal, M. M. Genetic and Non-Genetic Mechanisms Underlying Cancer Evolution. Can-cers (Basel) 2021, 13 (6). DOI: 10.3390/cancers13061380
(155) Morrison, S. J.; Kimble, J. Asymmetric and symmetric stem-cell divisions in development and cancer. Nature 2006, 441 (7097), 1068-1074. DOI: 10.1038/nature04956; Li, L.; Clevers, H. Coexistence of quiescent and active adult stem cells in mammals. Science 2010, 327 (5965), 542-545. DOI: 10.1126/science.1180794.).
(156) Lang, D.; Mascarenhas, J. B.; Shea, C. R. Melanocytes, melanocyte stem cells, and melanoma stem cells. Clin Derma-tol 2013, 31 (2), 166-178. DOI: 10.1016/j.clindermatol.2012.08.014.].
(157) Cabrera, M. C.; Hollingsworth, R. E.; Hurt, E. M. Cancer stem cell plasticity and tumor hierarchy. World J Stem Cells 2015, 7 (1), 27-36. DOI: 10.4252/wjsc.v7.i1.27.] has been added
Second paragraph
See lines: 352-370; the following paragraph [The involvement of the stem cell marker CD133 protein in the maintenance of melanoma stemness properties and drug resistance is mediated by its C-terminal do-main, which contains tyrosine binding sites located on tyrosine 828 (Tyr828) and tyrosine 852 (Tyr852) residues [74, 176]. These two tyrosine residues are phosphorylation targets of the non-receptor tyrosine kinase (NRTK) Fyn [176]. The contribution of CD133 to the regulation of CSCs functions such as self-renewal, differentiation, and drug resistance are likely mediated by the NRTK, Fyn-dependent mechanism via the phosphorylation of Tyr828 residue located on the cytoplasmic domain of CD133 [74, 176]. Our laboratory has demonstrated that the phosphorylation of Tyr828 is essential to trigger the activation of PI3K and its downstream dependent signaling pathways in melanoma [74]. Many studies have demonstrated that increased CD133 expression is associated with high tumorigenicity and metastatic potential for melanoma cells [75, 177- 179]. Also, CD133 protein has been implicated in the regulation of tumor re-sistance [180-182].
CD133-expressing CSCs have been shown to exhibit resistance to chemotherapy and radiation therapy in addition to being associated with poor prognosis in various cancers [182]. We and others demonstrated that CD133+ cancer cells confer resistance to many chemotherapeutic agents such as Caffeic acid phenethyl ester [14], Taxol [183], and fotemustine [74]. Accordingly, CD133-dependent mechanisms have been shown to be involved in the development of melanoma resistance to chemotherapy [74].] has been added.
References section
See lines: 1090-1113; the following references:
- Boivin, D.; Labbé, D.; Fontaine, N.; Lamy, S.; Beaulieu, E.; Gingras, D.; Béliveau, R. The stem cell marker CD133 (prominin-1) is phosphorylated on cytoplasmic tyrosine-828 and tyrosine-852 by Src and Fyn tyrosine kinases. Biochemistry 2009, 48(18), 3998-4007. DOI: 10.1021/bi900159d
- González-Herrero, I.; Romero-Camarero, I.; Cañueto, J.; Cardeñoso-Álvarez, E.; Fernández-López, E.; Pérez-Losada, J.; Sánchez-García, I.; Román-Curto, C. CD133+ cell content correlates with tumour growth in melanomas from skin with chronic sun-induced damage. Br J Dermatol 2013, 169 (4), 830-837. DOI: 10.1111/bjd.12428.
- Madjd, Z.; Erfani, E.; Gheytanchi, E.; Moradi-Lakeh, M.; Shariftabrizi, A.; Asadi-Lari, M. Expression of CD133 cancer stem cell marker in melanoma: a systematic review and meta-analysis. Int J Biol Markers 2016, 31 (2), e118-125. DOI: 10.5301/jbm.5000209.
- Liou, G. Y. CD133 as a regulator of cancer metastasis through the cancer stem cells. Int J Biochem Cell Biol 2019, 106, 1-7. DOI: 10.1016/j.biocel.2018.10.013.
- Lai, I. C.; Shih, P. H.; Yao, C. J.; Yeh, C. T.; Wang-Peng, J.; Lui, T. N.; Chuang, S. E.; Hu, T. S.; Lai, T. Y.; Lai, G. M. Elimination of cancer stem-like cells and potentiation of temozolomide sensitivity by Honokiol in glioblastoma multiforme cells. PLoS One 2015, 10 (3), e0114830. DOI: 10.1371/journal.pone.0114830.
- Yang, Y. P.; Chien, Y.; Chiou, G. Y.; Cherng, J. Y.; Wang, M. L.; Lo, W. L.; Chang, Y. L.; Huang, P. I.; Chen, Y. W.; Shih, Y. H.; et al. Inhibition of cancer stem cell-like properties and reduced chemoradioresistance of glioblastoma using microRNA145 with cationic polyurethane-short branch PEI. Biomaterials 2012, 33 (5), 1462-1476. DOI: 10.1016/j.biomaterials.2011.10.071.
- Tseng, L. M.; Huang, P. I.; Chen, Y. R.; Chen, Y. C.; Chou, Y. C.; Chen, Y. W.; Chang, Y. L.; Hsu, H. S.; Lan, Y. T.; Chen, K. H.; et al. Targeting signal transducer and activator of transcription 3 pathway by cucurbitacin I diminishes self-renewing and radiochemoresistant abilities in thyroid cancer-derived CD133+ cells. J Pharmacol Exp Ther 2012, 341 (2), 410-423. DOI: 10.1124/jpet.111.188730.
- Barzegar Behrooz, A.; Syahir, A.; Ahmad, S. CD133: beyond a cancer stem cell biomarker. J Drug Target 2019, 27 (3), 257-269. DOI: 10.1080/1061186X.2018.1479756.

Reviewer 2 Report
Comments and Suggestions for Authors
In this review, authors evaluated the role of cancer stem-like cells (CSCs) into the development of melanoma progression and drug resistance to the BRAF inhibitors in BRAF-mutated melanoma patients. The main biomarkers identifying the CSCs and, mostly, the pathways by which melanoma stem cells are generated were presented.
The manuscript includes all information and references regarding the role of CSCs; moreover, the Figures are highly explicative for readers.
Minor criticisms:
- more references about the molecular pathways involved in melanoma pathogenesis (considering those from NGS-based studies) should be included;
- some references are wrongly repeated (i.e Sensi et al. #31, #37, and #61; Yancovitz et al. #24 and #33; Flach et al. #229 and #230; Cuollo et al. #243 and #248)
Author Response
Editor
Cancers
Dear Editor,
Thank you very much for the encouraging comment regarding our Manuscript ID: cancers-2784052 “ Mechanisms of Melanoma Progression and Treatment Resistance: Role of
Cancer Stem-Like Cells .”
Enclosed find please our Point-for-Point response to the valuable comment of reviewer 2.
On behalf all my coauthors
Reviewer 2
Comments and Suggestions for Authors
In this review, authors evaluated the role of cancer stem-like cells (CSCs) in the development of melanoma progression and drug resistance to the BRAF inhibitors in BRAF-mutated melanoma patients. The main biomarkers identifying the CSCs and, mostly, the pathways by which melanoma stem cells are generated were presented.
The manuscript includes all information and references regarding the role of CSCs; moreover, the Figures are highly explicative for readers.
Minor criticisms:
Comment: - more references about the molecular pathways involved in melanoma pathogenesis (considering those from NGS-based studies) should be included;
Authors’response:
Thank you very much for your comment. Accordingly, we added more references reporting from NGS-based studies in melanoma pathogenesis and modified the section of Mechanisms of Tumor progression and drug resistance.
Section of Mechanisms of Tumor progression and drug resistance.
See Lines: 199-208; the following text [ The analysis of melanoma circulated tumor DNA (ctDNA) using next generation sequences (NGS) has been used as a reliable tool to monitor a driver mutation as predictor marker for survival high-risk stage III cutaneous melanoma patients [89, 90] and drug resistance [91]. The most mutated genes, which are associated with the development melanoma resistance to targeted therapy in melanoma include CDKN2A [92, 93], RB1 [92, 94], PIK3CA, AKT3, HOXD8 [95], PAX5 [93], MAP3K8 [96], and MITF [95]. All these genes are either involved in the regulation of MAPK and PI3K/AKT signaling pathways in addition to serving as tumor suppressors affecting drug resistance in other cancers including Homeobox protein Hox-D8 (HOXD8) [97] and Paired Box 5 (PAX5) [98].] has been added to the section of mechanisms of tumor progression and drug resistance.
References section
See lines:864-893; the following references [
(89) Diefenbach, R. J.; Lee, J. H.; Rizos, H. Monitoring Melanoma Using Circulating Free DNA. Am J Clin Dermatol 2019, 20 (1), 1-12. DOI: 10.1007/s40257-018-0398-x.
(90) Lee, J. H.; Saw, R. P.; Thompson, J. F.; Lo, S.; Spillane, A. J.; Shannon, K. F.; Stretch, J. R.; Howle, J.; Menzies, A. M.; Carlino, M. S.; et al. Pre-operative ctDNA predicts survival in high-risk stage III cutaneous melanoma patients. Ann Oncol 2019, 30 (5), 815-822. DOI: 10.1093/annonc/mdz075.
(91) Olbryt, M.; Pigłowski, W.; Rajczykowski, M.; Pfeifer, A.; Student, S.; Fiszer-Kierzkowska, A. Genetic Profiling of Advanced Melanoma: Candidate Mutations for Predicting Sensitivity and Resistance to Targeted Therapy. Target Oncol 2020, 15 (1), 101-113. DOI: 10.1007/s11523-020-00695-0.
(92) Long, G. V.; Fung, C.; Menzies, A. M.; Pupo, G. M.; Carlino, M. S.; Hyman, J.; Shahheydari, H.; Tembe, V.; Thompson, J. F.; Saw, R. P.; et al. Increased MAPK reactivation in early resistance to dabrafenib/trametinib combination therapy of BRAF-mutant metastatic melanoma. Nat Commun 2014, 5, 5694. DOI: 10.1038/ncomms6694
(93) Wheler, J.; Yelensky, R.; Falchook, G.; Kim, K. B.; Hwu, P.; Tsimberidou, A. M.; Stephens, P. J.; Hong, D.; Cronin, M. T.; Kurzrock, R. Next generation sequencing of exceptional responders with BRAF-mutant melanoma: implications for sensitivity and resistance. BMC Cancer 2015, 15, 61. DOI: 10.1186/s12885-015-1029-z.
(94) LoRusso, P. M.; Boerner, S. A.; Pilat, M. J.; Forman, K. M.; Zuccaro, C. Y.; Kiefer, J. A.; Liang, W. S.; Hunsberger, S.; Redman, B. G.; Markovic, S. N.; et al. Pilot Trial of Selecting Molecularly Guided Therapy for Patients with Non-V600 BRAF-Mutant Metastatic Melanoma: Experience of the SU2C/MRA Melanoma Dream Team. Mol Cancer Ther 2015, 14 (8), 1962-1971. DOI: 10.1158/1535-7163.MCT-15-0153
(95) Van Allen, E. M.; Wagle, N.; Sucker, A.; Treacy, D. J.; Johannessen, C. M.; Goetz, E. M.; Place, C. S.; Taylor-Weiner, A.; Whittaker, S.; Kryukov, G. V.; et al. The genetic landscape of clinical resistance to RAF inhibition in metastatic melanoma. Cancer Discov 2014, 4 (1), 94-109. DOI: 10.1158/2159-8290.CD-13-0617.
(96) Johannessen, C. M.; Boehm, J. S.; Kim, S. Y.; Thomas, S. R.; Wardwell, L.; Johnson, L. A.; Emery, C. M.; Stransky, N.; Cogdill, A. P.; Barretina, J.; et al. COT drives resistance to RAF inhibition through MAP kinase pathway reactivation. Nature 2010, 468 (7326), 968-972. DOI: 10.1038/nature09627.
(97) Sun, P.; Song, Y.; Liu, D.; Liu, G.; Mao, X.; Dong, B.; Braicu, E. I.; Sehouli, J. Potential role of the HOXD8 transcription factor in cisplatin resistance and tumour metastasis in advanced epithelial ovarian cancer. Sci Rep 2018, 8 (1), 13483. DOI: 10.1038/s41598-018-31030-3.
(98) Kurimoto, K.; Hayashi, M.; Guerrero-Preston, R.; Koike, M.; Kanda, M.; Hirabayashi, S.; Tanabe, H.; Takano, N.; Iwata, N.; Niwa, Y.; et al. PAX5 gene as a novel methylation marker that predicts both clinical outcome and cisplatin sensitivity in esophageal squamous cell carcinoma. Epigenetics 2017, 12 (10), 865-874. DOI: 10.1080/15592294.2017.1365207.] have been added to the references section.
Comment: - some references are wrongly repeated (i.e Sensi et al. #31, #37, and #61; Yancovitz et al. #24 and #33; Flach et al. #229 and #230; Cuollo et al. #243 and #248)
Authors’response: Thank you very much for your comment. Accordingly, we proved the references and removed and replaced the duplicated references with new ones.

Reviewer 3 Report
Comments and Suggestions for Authors
In this manuscript, Al Hmada et al. summarize the roles of melanoma stem-like cells (MSCs) in melanoma progression and drug resistance. After a short introduction, the authors begin with a discussion of the principles by which MSCs confer tissue heterogeneity and plasticity to melanoma, and then go on to describe in detail the molecular mechanisms of melanoma progression and drug resistance, drawing on an extensive literature review. The origin of cancer stem cells in paragraph 3 (Cancer stem cells) is a matter of debate, but in this manuscript it serves as an introduction to paragraph 4 (Mechanism of melanoma treatment failure and recurrence) and is described minimally. Paragraph 4 describes the mechanism of MSCs causing treatment resistance from multiple perspectives. In particular, I think it is good point that the SASP association in melanoma biology is described. Overall, I believe that this manuscript will not only be of interest to all readers involved in future research on MSCs, but will also be thought-provoking for readers involved in research on other cancer types. I have no objection to the acceptance of this manuscript as it is, although I cannot judge the English rendering.
Author Response
Editor
Cancers
Dear Editor,
Thank you very much for the encouraging comment regarding our Manuscript ID: cancers-2784052 “ Mechanisms of Melanoma Progression and Treatment Resistance: Role of
Cancer Stem-Like Cells .”
Enclosed find please our Point-for-Point response to the valuable comment of reviewer 3.
On behalf all my coauthors
Reviewer 3
Comments and Suggestions for Authors
In this manuscript, Al Hmada et al. summarize the roles of melanoma stem-like cells (MSCs) in melanoma progression and drug resistance. After a short introduction, the authors begin with a discussion of the principles by which MSCs confer tissue heterogeneity and plasticity to melanoma, and then go on to describe in detail the molecular mechanisms of melanoma progression and drug resistance, drawing on an extensive literature review. The origin of cancer stem cells in paragraph 3 (Cancer stem cells) is a matter of debate, but in this manuscript it serves as an introduction to paragraph 4 (Mechanism of melanoma treatment failure and recurrence) and is described minimally. Paragraph 4 describes the mechanism of MSCs causing treatment resistance from multiple perspectives. In particular, I think it is good point that the SASP association in melanoma biology is described. Overall, I believe that this manuscript will not only be of interest to all readers involved in future research on MSCs, but will also be thought-provoking for readers involved in research on other cancer types. I have no objection to the acceptance of this manuscript as it is, although I cannot judge the English rendering.
Authors ’response: Thank you very much for your comment.

Round 2
Reviewer 1 Report
Comments and Suggestions for Authors
The author answered the questions.